# Post-processing Private Synthetic Data
# for Improving Utility on Selected Measures

**Hao Wang, Shivchander Sudalairaj, John Henning,**
**Kristjan Greenewald, Akash Srivastava**[*]
MIT-IBM Watson AI Lab

## Abstract

Existing private synthetic data generation algorithms are agnostic to downstream tasks. However, end users may have specific requirements that the synthetic data must satisfy. Failure to meet these requirements could significantly reduce the utility of the data for downstream use. We introduce a post-processing technique that improves the utility of the synthetic data with respect to measures selected by the end user, while preserving strong privacy guarantees and dataset quality. Our technique involves resampling from the synthetic data to filter out samples that do not meet the selected utility measures, using an efficient stochastic first-order algorithm to find optimal resampling weights. Through comprehensive numerical experiments, we demonstrate that our approach consistently improves the utility of synthetic data across multiple benchmark datasets and state-of-the-art synthetic data generation algorithms.

## 1   Introduction

The advancement of machine learning (ML) techniques relies on large amounts of training data. However, data collection also poses a significant risk of exposing private information. In recent years, several instances of privacy breaches have surfaced [NS06, Con18], making it urgent to find a reliable way to share data. Today, the de facto standard for privacy protection is differential privacy (DP) [DR14]. DP ensures that useful information of a private dataset can be released while simultaneously preventing adversaries from identifying individuals' personal data. DP has been utilized effectively in a wide variety of settings, by actors including the US Census [Abo18] and various large corporations [App17, Fac20, RE19, HBMAL19, IBM23].

Generating private synthetic data is a crucial application of DP. It allows data scientists to train their ML models on the synthetic data while preserving a certain level of utility when deploying these models on real test data. The U.S. National Institute of Standards and Technology (NIST) has emphasized its significance by hosting a series of competitions [RTMT21, MMS21]. There is also significant work introducing new DP synthetic data generation mechanisms, including GAN-based [XLW+18, BJWW+19, JYVDS19, TWB+19], marginal-based [ZCP+17, MSM19, MMS21], and workload-based [VTB+20, ABK+21, LVW21, MMSM22, VAA+22] methods. Existing methods for generating private synthetic data are task-agnostic—they do not take into account the downstream use cases of the synthetic data in the data generation process. However, end users often have specific requirements for synthetic datasets to be successfully analyzed by their existing data science pipelines, which are often well-established, well-understood, heavily vetted, and difficult or expensive to change. Unfortunately, synthetic data generated by existing methods may not always meet these requirements, resulting in reduced utility for their downstream use cases. This raises a fundamental question:

---

[*]email: {hao, shiv.sr, john.l.henning, kristjan.h.greenewald, akash.srivastava}@ibm.com

*Is it possible to improve a synthetic dataset's utility on a set of selected measures while preserving its differential privacy guarantees?*

In this paper, we introduce a post-processing procedure that enhances the quality of the synthetic data on a set of utility measures. These measures are usually based on the end user's requirements for the synthetic data. Our method solves an optimization problem to calculate the optimal resampling weights for each data point. We then use these weights to resample the synthetic dataset, eliminating data points that do not accurately reflect the real data based on the selected utility measures. This post-processing procedure has many remarkable properties. Firstly, it is model-agnostic, meaning that it does not depend on any assumptions about how the synthetic data were generated. The synthetic data could be generated using either GAN-based, marginal-based, or workload-based methods. Second, the post-processed data preserve the DP guarantees while maintaining the performance of downstream models on real data compared to models trained on the original synthetic data. Finally, our algorithm is highly scalable, especially for high-dimensional data. It only requires a convex optimization whose number of variables is equal to the number of desired utility measures. We develop a stochastic first-order method that can efficiently solve this optimization problem. As a reference, we apply our algorithm to the `home-credit` dataset [MOK18] which consists of 307,511 data points and 104 features, and it only takes around 4 mins to finish running.

We conduct comprehensive numerical experiments to demonstrate the effectiveness of our post-processing techniques. Our results show that this algorithm consistently improves the utility of the synthetic data on different selected measures, across multiple datasets, and for synthetic data generated using various state-of-the-art privacy mechanisms. Additionally, the downstream models trained on the post-processed data can achieve comparable (or even better) performance on real data compared to the models trained on the original synthetic data. We demonstrate an application of our technique by using it to align the correlation matrix of the synthetic data with that of the real data. The correlation matrix shows the correlation coefficients between pairs of features and is often used as a reference for feature selection. By applying our post-processing technique, we can ensure that the correlation matrix of the synthetic data closely matches that of the real data. This, for instance, can help downstream data scientists more accurately select a task-appropriate set of features for fitting their ML models, e.g. by thresholding the correlation with the desired target variable or using Lasso techniques.

Our proof techniques rely on an information-theoretic formulation known as information projection, which dates back to Csiszár's seminal work [Csi75]. This formulation has recently been successfully applied in several domains, such as large deviation theory [DZ96], hypothesis testing [Csi84], and fair machine learning [AHJ+22]. Here we use information projection to "project" the empirical distribution of the synthetic data onto a set of distributions whose utility measures match those of the real data. We show that the optimal projected distribution can be expressed as an exponential tilting of the original distribution and determine the tilting parameters via a convex program in the dual space. To solve this optimization problem, we introduce a compositional proximal gradient algorithm, which is a stochastic first-order method and highly scalable for large dataset. Notably, this exponential tilting takes the form of a simple reweighting of the data points in the original synthetic dataset. Hence, our approach has the added benefit that it only reweights synthetic data points without changing their values, ensuring that the quality of the data points is not compromised and the support set of the overall dataset is not expanded.

In summary, our main contributions are:

- We present a post-processing technique that enables end users to customize synthetic data according to their specific requirements.
- Our technique is model-agnostic—it is applicable regardless of the methods used to generate private synthetic data. It is scalable to high-dimensional data and maintains strong privacy guarantees.
- We conduct comprehensive numerical experiments, showing that our technique can consistently improve the utility of synthetic data across multiple benchmark datasets and for synthetic data generated using state-of-the-art DP mechanisms.

The supplementary material of this paper includes: (i) omitted proofs of all theoretical results, (ii) details on algorithm implementation, and (iii) supporting experimental results.

## 1.1 Related Work

**DP synthetic data generation mechanisms.** Generating synthetic data using DP mechanisms is an active area of research [see e.g., HLM12, BLR13, GAH+14, CXZX15, BSG17, AZK+19, UV20, GMHI20, TMH+21, VAA+22, BSV22]. Along this line of work, the closest ones to ours are workload-aware methods [VTB+20, ABK+21, MMSM22, LVW21], which aim to ensure that the synthetic data perform well on a collection of queries. Our work differs in three key features. First, all existing approaches focus on generating private synthetic data from scratch, while we aim to post-process synthetic data to downweight samples that do not accurately represent the real data under the specified utility measures. As a result, our approach is more efficient as it does not need to fit a graphical model [MMSM22] or a neural network [LVW21] to represent the data distribution, and it is more extendable, allowing a single synthetic dataset to be quickly post-processed multiple times for different sets of utility measures as needed by downstream users. Second, some existing work may not scale well to large datasets as they either require solving an integer program multiple times [VTB+20] or need to solve a large-scale optimization problem [ABK+21]. In contrast, our approach is highly scalable, as it only requires solving a convex program whose number of variables is equal to the number of specified utility measures. Third, existing work often evaluates the quality of the synthetic data by how well it preserves key statistics (e.g., 3-way marginals) of the real data. In contrast, in our experiments, we evaluate our approach on the more stringent and realistic test of training various downstream ML models on the synthetic data and measuring their performance on real test data. The experimental results demonstrate that our approach can enhance the utility of synthetic data on selected measures without compromising their downstream quality.

**Public data assisted methods/Post-processing methods.** Our work introduces a post-processing procedure for improving the utility of a given synthetic dataset based on selected measures. As a special case, our approach can be applied to post-process publicly available datasets. In this regard, this work is related to public-data-assisted methods [see e.g., LVS+21, LVW21], which leverage public data for saving privacy budgets. We extend Algorithm 1 in [LVS+21] by formalizing the preservation of (noisy) utility measures as a constrained optimization problem, rather than minimizing the corresponding Lagrangian function. We further establish strong duality and propose a stochastic first-order method for solving this constrained optimization efficiently. We extend Algorithm 4 in [LVW21] by allowing any non-negative violation tolerance (i.e., any $\gamma \geq 0$ in (3b)) compared with $\gamma = 0$ in [LVW21]. This extension offers increased flexibility, as users can now select various values of $\gamma$ to navigate the trade-off between minimizing the distortion of the synthetic data and enhancing their utility on selected measures. Moreover, our experiments show that setting $\gamma$ to be a small positive number (e.g., $\gamma = 1e - 5$) consistently outperforms when $\gamma = 0$. Finally, our work is related with [NWD20], which proposes to post-process outputs from differentially private GAN-based models for improving the quality of the generated synthetic data. However, their method is tailored to GAN-based privacy mechanisms while our approach is model-agnostic. This versatility is crucial, given that marginal-based and workload-based mechanisms often yield higher quality synthetic tabular data, as evidenced by benchmark experiments in [TMH+21]. Our experiments indicate that our method consistently improves the utility of synthetic data produced by all kinds of privacy mechanisms, even when the initial synthetic data are of high quality.

## 2 Preliminaries and Problem Formulation

In this section, we review differential privacy and provide an overview of our problem setup.

### 2.1 Differential Privacy

We first recall the definition of differential privacy (DP) [DR14].

**Definition 1.** A randomized mechanism $\mathcal{M} : \mathcal{X}^n \to \mathcal{R}$ satisfies $(\epsilon, \delta)$-differential privacy, if for any adjacent datasets $\mathcal{D}$ and $\mathcal{D}'$, which only differ in one individual's record, and all possible outputs from the mechanism $\mathcal{O} \subseteq \mathcal{R}$, we have

$$\Pr(\mathcal{M}(\mathcal{D}) \in \mathcal{O}) \leq e^\epsilon \Pr(\mathcal{M}(\mathcal{D}') \in \mathcal{O}) + \delta. \tag{1}$$

DP has two important properties: post-processing immunity and composition rule. Specifically, if $\mathcal{M} : \mathcal{X}^n \to \mathcal{R}$ satisfies $(\epsilon, \delta)$-DP and $g : \mathcal{R} \to \mathcal{R}'$ is any randomized (or deterministic) function, then

$g \circ \mathcal{M} : \mathcal{X}^n \to \mathcal{R}'$ is $(\epsilon, \delta)$-DP; if $\mathcal{M} = (\mathcal{M}_1, \cdots, \mathcal{M}_k)$ is a sequence of randomized mechanisms, where $\mathcal{M}_i$ is $(\epsilon_i, \delta_i)$-DP, then $\mathcal{M}$ is $(\sum_{i=1}^{k} \epsilon_i, \sum_{i=1}^{k} \delta_i)$-DP. There are also advanced composition theorems [see e.g., DRV10, KOV15].

The Laplace and Gaussian mechanisms are two fundamental DP mechanisms. For a function $f : \mathcal{X}^n \to \mathbb{R}^K$, we define its $L_p$ sensitivity for $p \in \{1, 2\}$ as

$$\Delta_p(f) \triangleq \sup_{\mathcal{D} \sim \mathcal{D}'} \|f(\mathcal{D}) - f(\mathcal{D}')\|_p, \tag{2}$$

where $\mathcal{D}$ and $\mathcal{D}'$ are adjacent datasets. The Laplace mechanism adds Laplace noise $\mathrm{N} \sim \mathsf{Lap}(\mathbf{0}, b\boldsymbol{I})$ to the output of $f$:

$$\mathcal{M}(\mathcal{D}) = f(\mathcal{D}) + \mathrm{N}.$$

In particular, if $b = \Delta_1(f)/\epsilon$, the Laplace mechanism satisfies $(\epsilon, 0)$-DP. Similarly, the Gaussian mechanism adds Gaussian noise $\mathrm{N} \sim N(\mathbf{0}, \sigma^2 \boldsymbol{I})$ to the output of $f$. We recall a result from [BW18] that establishes a DP guarantee for the Gaussian mechanism and refer the reader to their Algorithm 1 for a way to compute $\sigma$ from $(\epsilon, \delta)$ and $\Delta_2(f)$.

**Lemma 1** ([BW18]). *For any $\epsilon \geq 0$ and $\delta \in [0, 1]$, the Gaussian mechanism is $(\epsilon, \delta)$-DP if and only if*

$$\Phi\left(\frac{\Delta_2(f)}{2\sigma} - \frac{\epsilon\sigma}{\Delta_2(f)}\right) - e^\epsilon \Phi\left(-\frac{\Delta_2(f)}{2\sigma} - \frac{\epsilon\sigma}{\Delta_2(f)}\right) \leq \delta,$$

*where $\Phi$ is the Gaussian CDF: $\Phi(t) \triangleq \frac{1}{\sqrt{2\pi}} \int_{-\infty}^{t} e^{-y^2/2} dy$.*

The function $f$ can often be determined by a query $q : \mathcal{X} \to \mathbb{R}$: $f(\mathcal{D}) = \frac{1}{n} \sum_{\boldsymbol{x}_i \in \mathcal{D}} q(\boldsymbol{x}_i)$. In this case, we denote $\Delta(q) \triangleq \sup_{x, x' \in \mathcal{X}} |q(x) - q(x')|$; $q(P) \triangleq \mathbb{E}_{\mathrm{X} \sim P}[q(\mathrm{X})]$ for a probability distribution $P$; and $q(\mathcal{D}) \triangleq \frac{1}{n} \sum_{\boldsymbol{x}_i \in \mathcal{D}} q(\boldsymbol{x}_i)$ for a dataset $\mathcal{D}$.

## 2.2 Problem Formulation

Consider a dataset $\mathcal{D}$ comprising private data from $n$ individuals, where each individual contributes a single record $\boldsymbol{x} = (x_1, \cdots, x_d) \in \mathcal{X}$. Suppose that a synthetic dataset $\mathcal{D}_{\text{syn}}$ has been generated from the real dataset $\mathcal{D}$, but it may not accurately reflect the real data on a set of utility measures (also known as workload). We express the utility measures of interest as a collection of queries, denoted by $\mathcal{Q} \triangleq \{q_k : \mathcal{X} \to \mathbb{R}\}_{k=1}^{K}$, and upper bound their $L_p$ sensitivity by $\Delta_p(\mathcal{Q}) \triangleq \frac{1}{n} (\sum_{k \in [K]} \Delta(q_k)^p)^{1/p}$.

Our goal is to enhance the utility of the synthetic dataset $\mathcal{D}_{\text{syn}}$ w.r.t. the utility measures $\mathcal{Q}$ while maintaining the DP guarantees. We do not make any assumption about how these synthetic data have been generated—e.g., they could be generated by either GAN-based, marginal-based, or workload-based methods. We evaluate the utility measures from the real data via a $(\epsilon_{\text{post}}, \delta_{\text{post}})$-DP mechanism and denote their answers by $\boldsymbol{a} = (a_1, \cdots, a_K)$. Next, we present a systematic approach to post-process the synthetic data, aligning their utility with the (noisy) answers $\boldsymbol{a}$.

## 3 Post-processing Private Synthetic Data

In this section, we present our main result: a post-processing procedure for improving the utility of a synthetic dataset. We formulate this problem as an optimization problem that projects the probability distribution of the synthetic data onto a set of distributions that align the utility measures with the (noisy) real data. We prove that the optimal projected distribution can be expressed in a closed form, which is an exponentially tilted distribution of the original synthetic data distribution. To compute the tilting parameters, we introduce a stochastic first-order method and leverage the tilting parameters to resample from the synthetic data. By doing so, we filter out synthetic data that inaccurately represent the real data on the selected set of utility measures.

We use $P_{\text{syn}}$ and $P_{\text{post}}$ to represent the (empirical) probability distribution of the synthetic data and the generating distribution of post-processed data, respectively. We obtain the post-processed distribution

by "projecting" $P_{\text{syn}}$ onto the set of distributions that align with the real data utility measures up to $\gamma \geq 0$:

$$\min_{P_{\text{post}}} \ D_{\text{KL}}(P_{\text{post}} \| P_{\text{syn}}), \tag{3a}$$

$$\text{s.t. } |q_k(P_{\text{post}}) - a_k| \leq \gamma, \quad \forall k \in [K]. \tag{3b}$$

Here the "distance" between two probability distributions is measured using the KL-divergence. Solving this problem directly suffers from the curse of dimensionality, as the number of variables grows exponentially with the number of features. Below, we leverage strong duality and prove that the optimal distribution $P_{\text{post}}^*$ can be achieved by tilting $P_{\text{syn}}$ and the tilting parameters can be obtained from a convex program. This approach involves fewer variables, making it more efficient and practical.

**Theorem 1.** *If* (3) *is feasible, then its optimal solution has a closed-form expression:*

$$P_{post}^*(\boldsymbol{x}) \propto P_{syn}(\boldsymbol{x}) \exp\left(-\sum_{k=1}^{K} \lambda_k^* \left(q_k(\boldsymbol{x}) - a_k\right)\right), \tag{4}$$

*where the dual variables* $\boldsymbol{\lambda}^* \triangleq (\lambda_1^*, \cdots, \lambda_K^*)$ *are the optimal solution of the following optimization:*

$$\min_{\boldsymbol{\lambda} \in \mathbb{R}^K} \ \underbrace{\log \mathbb{E}_{X \sim P_{syn}}\left[\exp\left(-\sum_{k=1}^{K} \lambda_k (q_k(X) - a_k)\right)\right]}_{\triangleq f(\boldsymbol{\lambda})} + \gamma \|\boldsymbol{\lambda}\|_1. \tag{5}$$

Theorem 1 only requires the feasibility of (3). To ensure feasibility, we denoise the (noisy) answers $\boldsymbol{a}$ computed from real data. This denoising step could also increase the accuracy of the released answers while preserving DP guarantees due to the post-processing property [BW18, LWK15]. Specifically, suppose that $\mathcal{D}_{\text{syn}}$ has $S$ unique values. We denote the collection of these unique values by $\text{supp}(\mathcal{D}_{\text{syn}}) = \{\boldsymbol{x}_1, \cdots, \boldsymbol{x}_S\}$. We construct a matrix $\boldsymbol{Q} \in \mathbb{R}^{K \times S}$ such that $Q_{j,s} = q_j(\boldsymbol{x}_s)$ and solve the following linear or quadratic programs (see Appendix B for more details):

$$\boldsymbol{p}^* = \begin{cases} \operatorname{argmin}_{\boldsymbol{p} \in \Delta_S^+} \ \|\boldsymbol{Q}\boldsymbol{p} - \boldsymbol{a}\|_1, & \text{if using Laplace mechanism,} \\ \operatorname{argmin}_{\boldsymbol{p} \in \Delta_S^+} \ \frac{1}{2}\|\boldsymbol{Q}\boldsymbol{p} - \boldsymbol{a}\|_2^2, & \text{if using Gaussian mechanism.} \end{cases}$$

Here $\Delta_S^+ = \{\boldsymbol{p} \in \mathbb{R}^S \mid \mathbf{1}^T \boldsymbol{p} = 1, \boldsymbol{p} > 0\}$ is the interior of the probability simplex in $\mathbb{R}^S$. We obtain the denoised answers as $\boldsymbol{a}^* = \boldsymbol{Q}\boldsymbol{p}^* = (a_1^*, \cdots, a_K^*)$. Finally, if we solve (3) with $\boldsymbol{a}^*$ instead, then it is guaranteed to be feasible.

Although (5) is a convex program, its form differs from the standard empirical risk minimization as the expected loss is composed with a logarithmic function. Thus, one cannot directly apply stochastic gradient descent (SGD) to solve this problem as computing an unbiased (sub)-gradient estimator for (5) is challenging. Instead, we apply a stochastic compositional proximal gradient algorithm to solve (5). Note that the partial derivative of $f$ in (5) has a closed-form expression:

$$\frac{\partial f}{\partial \lambda_j}(\boldsymbol{\lambda}) = -\frac{\mathbb{E}_{X \sim P_{\text{syn}}}\left[\exp\left(-\sum_{k=1}^{K} \lambda_k(q_k(X) - a_k)\right) \cdot (q_j(X) - a_j)\right]}{\mathbb{E}_{X \sim P_{\text{syn}}}\left[\exp\left(-\sum_{k=1}^{K} \lambda_k(q_k(X) - a_k)\right)\right]}.$$

We estimate this partial derivative by using a mini-batch of data to calculate the expectations in numerator and denominator separately, and then computing their ratio. With this estimator, we can apply the stochastic proximal gradient algorithm [PB14] to solve (5). Although this approach uses a biased gradient estimator, it has favorable convergence guarantees [see e.g., WLF16, HZCH20]. We provide more details about the updating rule in Algorithm 1.

After obtaining the optimal dual variables from Algorithm 1, we generate the post-processed dataset by resampling from the synthetic data with weights: $\exp(-\sum_{k=1}^{K} \lambda_k^* (q_k(\boldsymbol{x}) - a_k))$ for each point $\boldsymbol{x}$. This ensures that the resampled data follow the optimal distribution $P_{\text{post}}^*$ in (4). As our approach only reweights synthetic data without altering their values, the quality of the data points is preserved and the support set of the overall dataset is not expanded. Moreover, our numerical experiments

**Algorithm 1** Dual variables computation.

---

**Input:** $\mathcal{D} = \{\bar{q}_i\}_{i=1}^n$ where $\bar{q}_i = (q_1(\boldsymbol{x}_i^{\text{syn}}) - a_1, \cdots, q_K(\boldsymbol{x}_i^{\text{syn}}) - a_K)$; maximal number of iterations $T$; mini-batch index $\mathcal{B}_t \subseteq [n]$; step size $\alpha_t$

**for** $t = 1, \cdots, T$ **do**

    Dual variables update:

$$\lambda_j^{(t+1)} = \text{prox}_{\alpha_t \gamma \|\cdot\|_1} \left( \lambda_j^{(t)} + \frac{\alpha_t}{\tau^{(t)}} \frac{1}{|\mathcal{B}_t|} \sum_{i \in \mathcal{B}_t} \exp\left( -\sum_{k=1}^K \lambda_k^{(t)} \bar{q}_{i,k} \right) \bar{q}_{i,j} \right) \quad \text{for } j \in [K]$$

$$\tau^{(t+1)} = \frac{1}{|\mathcal{B}_t|} \sum_{i \in \mathcal{B}_t} \exp\left( -\sum_{k=1}^K \lambda_k^{(t+1)} \bar{q}_{i,k} \right)$$

**end for**
**Output:** $\boldsymbol{\lambda}^{(T+1)}$

---

**Function:** $\text{prox}_{\alpha\|\cdot\|_1}(x) = \text{sign}(x) \max\{|x| - \alpha, 0\}$

---

demonstrate that downstream predictive models trained on the post-processed synthetic data can achieve comparable or even better performance on real data than those trained on the original synthetic data.

Finally, we establish a DP guarantee for the post-processing data. Since our algorithm only accesses the real data when computing the utility measures, the composition rule yields the following proposition.

**Proposition 1.** *Suppose the synthetic data are generated from any $(\epsilon, \delta)$-DP mechanism and the utility measures are evaluated from the real data from any $(\epsilon_{post}, \delta_{post})$-DP mechanism. Then the post-processed data satisfy $(\epsilon + \epsilon_{post}, \delta + \delta_{post})$-DP.*

So far, we have introduced a general framework for post-processing synthetic data to improve their utility on selected measures. Next, we present a concrete use case of our framework: aligning the correlation matrix. This can be achieved by choosing a specific set of utility measures in our framework.

**Aligning the correlation matrix.** The correlation matrix comprises the correlation coefficients between pairs of features and is commonly used by data scientists to select features before fitting their predictive models. Aligning the correlation matrix of synthetic data with real data can assist data scientists in identifying the most suitable set of features and developing a more accurate model. Recall the definition of the correlation coefficient between features $X_i$ and $X_j$ for $i, j \in [d]$:

$$\text{corr}(X_i, X_j) = \frac{\mathbb{E}[X_i X_j] - \mathbb{E}[X_i]\mathbb{E}[X_j]}{\sqrt{\mathbb{E}[X_i^2] - \mathbb{E}[X_i]^2} \cdot \sqrt{\mathbb{E}[X_j^2] - \mathbb{E}[X_j]^2}}.$$

Since the correlation coefficient only depends on the first and second order moments, we can select the following set of $(d+3)d/2$ queries as our utility measures:

$$\mathcal{Q} = \{q_i(\boldsymbol{x}) = x_i\}_{i \in [d]} \cup \{q_{i,j}(\boldsymbol{x}) = x_i x_j\}_{i \leq j}.$$

Assume that each feature $x_i$ is normalized and takes values within $[0, 1]$. As a result, the $L_1$ and $L_2$ sensitivity of $\mathcal{Q}$ can be upper bounded:

$$\Delta_1(\mathcal{Q}) \leq \frac{(d+3)d}{2n} \quad \text{and} \quad \Delta_2(\mathcal{Q}) \leq \frac{\sqrt{(d+3)d}}{\sqrt{2}n}.$$

We can compute these utility measures using the Laplace or Gaussian mechanism, denoise the answers, compute the optimal dual variables, and resample from the synthetic data.

## 4 Numerical Experiments

We now present numerical experiments to demonstrate the effectiveness of our post-processing algorithm. Our results show that this algorithm consistently improves the utility of the synthetic

data across multiple datasets and state-of-the-art privacy mechanisms. Additionally, it achieves this without degrading the performance of downstream models or statistical metrics. We provide additional experimental results and implementation details in Appendix C.

**Benchmark.** We evaluate our algorithm on four benchmark datasets: `Adult`, `Bank`, `Mushroom`, and `Shopping`, which are from the UCI machine learning repository [DG17]. All the datasets have a target variable that can be used for a downstream prediction task. We use four existing DP mechanisms—`AIM` [MMSM22], `MST` [MMS21], `DPCTGAN` [RLP+20], and `PATECTGAN` [RLP+20]— to generate private synthetic data. Among these DP mechanisms, `AIM` is a workload-based method; `MST` is a marginal-based method; `DPCTGAN` and `PATECTGAN` are GAN-based methods. All of their implementations are from the OpenDP library [Sma23]. To demonstrate the scalability of our approach, we conduct an additional experiment on the `home-credit` dataset [MOK18]. This dataset has 307,511 data points and 104 features. We apply `GEM` [LVW21] to this high-dimensional dataset for generating synthetic data. We pre-process each dataset to convert categorical columns into numerical features and standardize all features to the range $[0, 1]$.

We clarify that our objective is not to compare the quality of synthetic data generated by these DP mechanisms (for a benchmark evaluation, please refer to [TMH+21]), but to demonstrate how our post-processing method can consistently improve the utility of synthetic data generated from these different DP mechanisms.

**Setup.** We compare synthetic data produced by different privacy mechanisms with $\epsilon \in \{2, 4\}$ against the post-processed synthetic data that are generated by applying our post-processing technique with $\epsilon_{\text{post}} = 1$ to synthetic data generated from privacy mechanisms with $\epsilon \in \{1, 3\}$. For UCI datasets, we select 5 features, including the target variable, from the synthetic data that have the highest absolute correlation with the target variable. These features are chosen as they have a higher influence on downstream prediction tasks. The set of these features is denoted by $\mathcal{S} \subseteq [d]$. Next, we define the utility measures as the first-order and second-order moment queries among the features in $\mathcal{S}$: $\mathcal{Q} = \{q_i(\boldsymbol{x}) = x_i\}_{i \in \mathcal{S}} \cup \{q_{i,j}(\boldsymbol{x}) = x_i x_j\}_{i,j \in \mathcal{S}}$. For `home-credit` dataset, we follow the same setup but choosing 10 features.

We apply the Gaussian mechanism with $(\epsilon_{\text{post}} = 1, \delta_{\text{post}} = 1/n^2)$ to estimate utility measures from the real data, where $n$ denotes the number of real data points. Finally, we apply Algorithm 1 with $\gamma = 1e-5$, a batch size of 256 for UCI datasets and 4096 for `home-credit`, and 200 epochs to compute the optimal resampling weights, which are then used to resample from the synthetic data with the same sample size. We repeat our experiment 5 times to obtain an error bar for each evaluation measure.

**Evaluation measure.** We provide evaluation metrics for assessing the quality of synthetic data after post-processing. Our first metric, *utility improvement*, is defined as:

$$1 - \frac{\text{error}_{\text{corr}}(\mathcal{D}_{\text{post}})}{\text{error}_{\text{corr}}(\mathcal{D}_{\text{syn}})}, \quad \text{where } \text{error}_{\text{corr}}(\mathcal{D}) = \|\text{corr}(\mathcal{D}_{\text{real}}) - \text{corr}(\mathcal{D})\|_1 \text{ for } \mathcal{D} \in \{\mathcal{D}_{\text{post}}, \mathcal{D}_{\text{syn}}\}.$$

The range of this metric is $(-\infty, 1]$, where a value of 1 indicates a perfect alignment between the correlation matrix of the post-processed data and the real data.

To assess the downstream quality of synthetic data, we train predictive models using the synthetic data and then test their performance on real data. Specifically, we split the real data, using 80% for generating synthetic data and setting aside 20% to evaluate the performance of predictive models. We use the synthetic data to train a logistic regression classifier to predict the target variable based on other features. Then we apply this classifier to real data and calculate its F1 score. We use the function BinaryLogisticRegression from `SDMetrics` library [Dat23] to implement this process.

We compute two statistical measures to quantify the similarity between synthetic and real data distributions. The first measure is the average Jensen-Shannon distance between the marginal distributions of synthetic and real data, which ranges from 0 (identical distribution) to 1 (totally different distributions). The second measure is the average inverse KL-divergence between the marginal distributions of synthetic and real data, which ranges from 0 (totally different distributions) to 1 (identical distribution). These statistical measures are implemented by using `synthcity` library [QCvdS23].

| Dataset | DP Mechanism | Utility Improv. | F1 score | | JS distance (marginal) | | Inverse KL (marginal) | |
|---|---|---|---|---|---|---|---|---|
| | | | w/o post-proc. | w/ post-proc. | w/o post-proc. | w/ post-proc. | w/o post-proc. | w/ post-proc. |
| Adult | AIM | **0.13** ±0.03 | 0.61 | 0.61 ±0.0 | 0.01 | 0 ±0.0 | 0.99 | 1 ±0.0 |
| | MST | **0.22** ±0.02 | 0.55 | 0.56 ±0.0 | 0.01 | 0 ±0.0 | 1 | 1 ±0.0 |
| | DPCTGAN | **0.81** ±0.09 | 0.22 | 0.33 ±0.02 | 0.07 | 0.03 ±0.02 | 0.85 | 0.91 ±0.0 |
| | PATECTGAN | **0.6** ±0.04 | 0.37 | 0.5 ±0.03 | 0.04 | 0.01 ±0.0 | 0.72 | 0.87 ±0.0 |
| Mushroom | AIM | **0.12** ±0.0 | 0.93 | 0.93 ±0.0 | 0.01 | 0 ±0.0 | 1 | 1 ±0.0 |
| | MST | **0.58** ±0.0 | 0.83 | 0.83 ±0.02 | 0.01 | 0 ±0.0 | 1 | 1 ±0.0 |
| | DPCTGAN | **0.69** ±0.04 | 0.47 | 0.68 ±0.01 | 0.05 | 0.02 ±0.0 | 0.83 | 0.92 ±0.0 |
| | PATECTGAN | **0.83** ±0.05 | 0.6 | 0.86 ±0.04 | 0.03 | 0.02 ±0.0 | 0.87 | 0.95 ±0.0 |
| Shopper | AIM | **0.1** ±0.02 | 0.48 | 0.48 ±0.02 | 0.02 | 0.01 | 0.84 | 0.92 ±0.0 |
| | MST | **0.5** ±0.02 | 0.42 | 0.47 ±0.02 | 0.01 | 0 ±0.0 | 0.99 | 1 ±0.0 |
| | DPCTGAN | **0.36** ±0.05 | 0.27 | 0.3 ±0.02 | 0.03 | 0.01 ±0.0 | 0.74 | 0.85 ±0.0 |
| | PATECTGAN | **0.11** ±0.04 | 0.25 | 0.31 ±0.05 | 0.04 | 0.01 ±0.0 | 0.8 | 0.89 ±0.0 |
| Bank | AIM | **0.18** ±0.01 | 0.45 | 0.46 ±0.01 | 0.04 | 0.02 ±0.01 | 0.83 | 0.9 ±0.0 |
| | MST | **0.32** ±0.02 | 0.43 | 0.44 ±0.02 | 0.02 | 0.01 ±0.0 | 0.98 | 1 ±0.0 |
| | DPCTGAN | **0.2** ±0.02 | 0.22 | 0.24 ±0.07 | 0.04 | 0.02 ±0.01 | 0.71 | 0.88 ±0.0 |
| | PATECTGAN | **0.25** ±0.03 | 0.2 | 0.23 ±0.05 | 0.03 | 0.01 ±0.0 | 0.83 | 0.9 ±0.0 |

Table 1: We compare synthetic data generated without and with our post-processing technique, all under the same privacy budget $\epsilon = 2$. We demonstrate utility improvement (higher is better, positive numbers imply improvement) and F1 score for downstream models trained on synthetic data and tested on real data. For reference, when training the same downstream model using real data, the F1 scores are: (Adult, $0.61$), (Mushroom, $0.95$), (Shopper, $0.54$), and (Bank, $0.47$). Additionally, we measure the average Jensen-Shannon (JS) distance between the marginal distributions of synthetic and real data (0: identical distribution; 1: totally different distributions) and the average inverse KL-divergence (0: totally different distributions; 1: identical distribution). As shown, our technique consistently improves the utility of the synthetic data across all datasets and all DP mechanisms without degrading the performance of downstream models or statistical metrics.

| DP Mechanism | Utility Improv. | F1 score w/o post-proc. | F1 score w/ post-proc. | JS distance w/o post-proc. | JS distance w/ post-proc. | Inverse KL w/o post-proc. | Inverse KL w/ post-proc. |
|---|---|---|---|---|---|---|---|
| $\text{GEM}_{\epsilon=2}$ | **0.57** ±0.03 | 0.19 | 0.19 ±0.02 | 0.01 | 0.01 ±0.0 | 0.93 | 0.97 ±0.01 |
| $\text{GEM}_{\epsilon=4}$ | **0.68** ±0.01 | 0.21 | 0.21 ±0.01 | 0.01 | 0.01 ±0.0 | 0.95 | 0.99 ±0.01 |

Table 2: Experimental results on the `home-credit` dataset. We compare synthetic data produced by GEM with $\epsilon \in \{2, 4\}$ against the post-processed synthetic data that are generated by applying our post-processing technique with $\epsilon_{\text{post}} = 1$ to synthetic data generated from GEM with $\epsilon \in \{1, 3\}$. For reference, the downstream model trained and tested on real data has an F1 score of $0.24$.

**Result.** We present the experimental results on the UCI datasets in Table 1 (and Table 3 in Appendix C with a higher privacy budget). We report *utility improvement*, *F1 score*, and *statistical measures*. As shown, our post-processing technique consistently enhances the utility of synthetic data on selected metrics across all benchmark datasets and all privacy mechanisms. Moreover, the utility improvements are achieved without degrading the performance of downstream models or statistical properties of synthetic data. In other words, our post-processing procedure ensures that the logistic regression classifier trained on synthetic data can achieve comparable or even higher performance on real test data, while simultaneously reducing or maintaining the statistical divergences between synthetic and real data.

We present the experimental results on the `home-credit` dataset in Table 2 and illustrate the misalignment of the correlation matrix with and without applying our post-processing procedure in Figure 1. The results demonstrate that our algorithm consistently reduces the overall correlation misalignment. Additionally, our procedure, which includes computing the utility measures from real data, denoising the noisy answers, and computing optimal resampling weights, only takes around 4 mins on 1x NVIDIA GeForce RTX 3090 GPU.

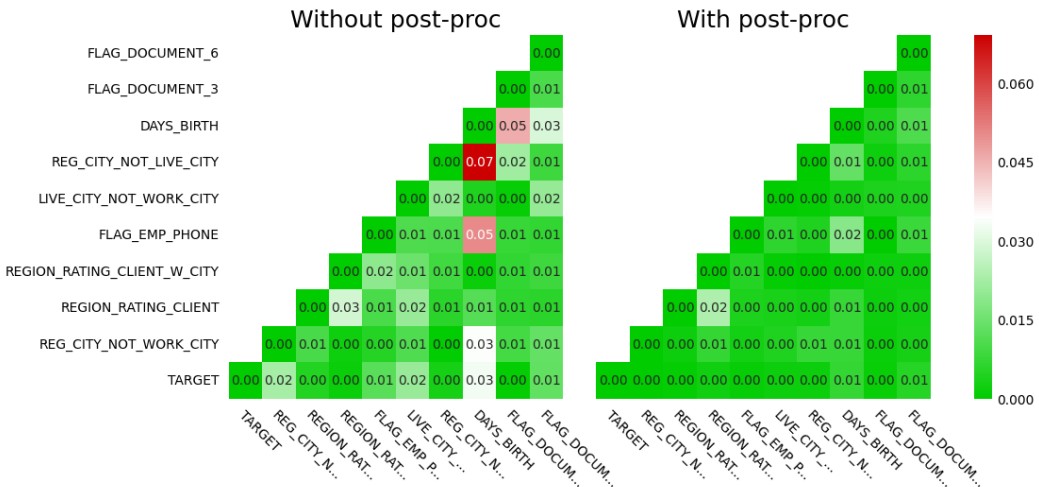

Figure 1: Correlation matrix *misalignment* with and without applying our post-processing procedure. Left: absolute difference between the real and *synthetic* data correlation matrices. Right: absolute difference between the real and *post-processed* synthetic data correlation matrices. As shown, our procedure effectively aligns the correlation matrix of the synthetic data with that of the real data.

# 5    Conclusion and Limitations

The use of private synthetic data allows for data sharing without compromising individuals' private information. In practice, end users typically have specific requirements that the synthetic data must meet, which are often derived from their standard data science pipeline. Failure to meet these requirements can diminish the usefulness of the synthetic data. To address this issue, we introduce a theoretically-grounded post-processing procedure for improving a synthetic dataset's utility on a selected set of measures. This procedure filters out samples that do not accurately reflect the real data, while still maintaining DP guarantees. We also present a scalable algorithm for computing resampling weights and demonstrate its effectiveness through extensive experiments.

We believe there is a crucial need for future research to comprehend the capabilities and limitations of private synthetic data from various perspectives [see e.g., JSH+22, GODC22, BDI+23, for further discussions]. It should be noted that synthetic data generation is not intended to replace real data, and any model trained on synthetic data must be thoroughly evaluated before deployment on real data. In cases a ML model trained on synthetic data exhibits algorithmic bias upon deployment on real data, it becomes imperative to identify the source of bias and repair the model by querying the real data with a privacy mechanism.

When generating synthetic data using DP mechanisms, outliers may receive higher noise, which can have a greater impact on minority groups. Consequently, ML models trained on DP-synthetic data by downstream users may demonstrate a disparate impact when deployed on real data [GODC22]. This is true even when using fairness-intervention algorithms [HCZ+22] during model training because the synthetic data follows a different probability distribution than real data. One can potentially apply our post-processing techniques to resample synthetic data and eliminate samples that do not reflect the underlying distribution of the population groups, while ensuring sample diversity. By doing so, they can achieve a higher fairness-accuracy curve when deploying downstream ML models trained on the post-processed synthetic data on real data.

Another promising avenue of research is to investigate the data structure in cases where a user or a company can provide multiple records to the dataset. Financial data, for instance, is typically collected over time and presented as a time series. While it is possible to combine all of a user's data into a single row, this approach can lead to an unmanageably unbounded feature domain and limit the effectiveness of the synthetic data. Therefore, it is worthwhile to explore other privacy notions [see e.g., ZXLZ14, DWCS19, LSA+21] and develop strategies for optimizing higher privacy-utility trade-offs under special data structures.

## Acknowledgement

H. Wang would like to thank Rui Gao (UT Austin) for his valuable input in the early stage of this project.

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

# A Omitted Proofs

## A.1 Proof of Theorem 1

Recall that Slater's condition is a sufficient condition for ensuring strong duality in the context of convex program [see Section 5.2.3 in BV04]: it states that strong duality holds for a convex program if there exists a strictly feasible solution (i.e., a solution that satisfies all constraints and ensures that nonlinear constraints are satisfied with strict inequalities).

*Proof.* Since (3) only has linear constraints and is feasible, Slater's condition holds. Recall that $P_{\mathrm{syn}}$ represents the empirical distribution of the synthetic data, supported on $\mathcal{X}$. Without loss of generality, we assume $\mathcal{X}$ is a finite set and denote $\boldsymbol{p}^{\mathrm{post}} \triangleq (P_{\mathrm{post}}(\boldsymbol{x}))_{\boldsymbol{x} \in \mathcal{X}}$ and $\boldsymbol{p}^{\mathrm{syn}} \triangleq (P_{\mathrm{syn}}(\boldsymbol{x}))_{\boldsymbol{x} \in \mathcal{X}}$ where $p_{\boldsymbol{x}}^{\mathrm{post}} \triangleq P_{\mathrm{post}}(\boldsymbol{x})$, $p_{\boldsymbol{x}}^{\mathrm{syn}} \triangleq P_{\mathrm{syn}}(\boldsymbol{x})$. For each query, we denote $\boldsymbol{q}_k \triangleq (q_k(\boldsymbol{x}))_{\boldsymbol{x} \in \mathcal{X}}$. Now we introduce the dual variables $\mu$, $\boldsymbol{\lambda}^+ \triangleq (\lambda_1^+, \cdots, \lambda_K^+)$, and $\boldsymbol{\lambda}^- \triangleq (\lambda_1^-, \cdots, \lambda_K^-)$. Then the Lagrangian function of (3) can be written as

$$L(\boldsymbol{p}^{\mathrm{post}}, \boldsymbol{\lambda}^+, \boldsymbol{\lambda}^-, \mu)$$

$$\triangleq \sum_{\boldsymbol{x} \in \mathcal{X}} p_{\boldsymbol{x}}^{\mathrm{post}} \log \frac{p_{\boldsymbol{x}}^{\mathrm{post}}}{p_{\boldsymbol{x}}^{\mathrm{syn}}} + \mu \left( \mathbf{1}^T \boldsymbol{p}^{\mathrm{post}} - 1 \right) + \sum_{k=1}^K \lambda_k^+ \left( \boldsymbol{q}_k^T \boldsymbol{p}^{\mathrm{post}} - a_k - \gamma \right) + \lambda_k^- \left( -\boldsymbol{q}_k^T \boldsymbol{p}^{\mathrm{post}} + a_k - \gamma \right).$$

The Lagrangian function is strictly convex w.r.t. $\boldsymbol{p}^{\mathrm{post}}$ so it has a unique solution if exists. We compute the partial derivative w.r.t. $\boldsymbol{p}^{\mathrm{post}}$ and set it to be zero:

$$\frac{\partial L(\boldsymbol{p}^{\mathrm{post}}, \boldsymbol{\lambda}^+, \boldsymbol{\lambda}^-, \mu)}{\partial p_{\boldsymbol{x}}^{\mathrm{post}}} = \log \frac{p_{\boldsymbol{x}}^{\mathrm{post}}}{p_{\boldsymbol{x}}^{\mathrm{syn}}} + 1 + \mu + \sum_{k=1}^K (\lambda_k^+ - \lambda_k^-) q_{k,\boldsymbol{x}} = 0. \tag{6}$$

We denote $\boldsymbol{\lambda} \triangleq \boldsymbol{\lambda}^{+,*} - \boldsymbol{\lambda}^{-,*}$. Then (6) yields a closed-form expression of the optimal solution:

$$p_{\boldsymbol{x}}^{\mathrm{post},*} = p_{\boldsymbol{x}}^{\mathrm{syn}} \exp \left( -\sum_{k=1}^K \lambda_k q_{k,\boldsymbol{x}} - 1 - \mu^* \right).$$

Substitute this optimal solution into the Lagrangian function and rewrite it as

$$L(\boldsymbol{p}^{\mathrm{post},*}, \boldsymbol{\lambda}^+, \boldsymbol{\lambda}^-, \mu)$$

$$= \sum_{\boldsymbol{x} \in \mathcal{X}} p_{\boldsymbol{x}}^{\mathrm{post},*} \left( -\sum_{k=1}^K \lambda_k q_{k,\boldsymbol{x}} - 1 - \mu \right) + \mu \left( \sum_{\boldsymbol{x} \in \mathcal{X}} p_{\boldsymbol{x}}^{\mathrm{post},*} - 1 \right) + \sum_{k=1}^K \sum_{\boldsymbol{x} \in \mathcal{X}} \lambda_k q_{k,\boldsymbol{x}} p_{\boldsymbol{x}}^{\mathrm{post},*} + \ldots$$

$$= -\sum_{\boldsymbol{x} \in \mathcal{X}} p_{\boldsymbol{x}}^{\mathrm{post},*} - \mu + \ldots$$

$$= -\sum_{\boldsymbol{x} \in \mathcal{X}} p_{\boldsymbol{x}}^{\mathrm{syn}} \exp \left( -\sum_{k=1}^K \lambda_k q_{k,\boldsymbol{x}} - 1 - \mu \right) - \mu + \ldots$$

where we omit the terms that do not depend on $\mu$. Therefore, the $\mu$ that maximizes $L(\boldsymbol{p}^{\mathrm{post},*}, \boldsymbol{\lambda}^+, \boldsymbol{\lambda}^-, \mu)$ satisfies:

$$\sum_{\boldsymbol{x} \in \mathcal{X}} p_{\boldsymbol{x}}^{\mathrm{syn}} \exp \left( -\sum_{k=1}^K \lambda_k q_{k,\boldsymbol{x}} - 1 - \mu^* \right) - 1 = 0.$$

We simplify the above expression and obtain

$$\exp \left( 1 + \mu^* \right) = \sum_{\boldsymbol{x} \in \mathcal{X}} p_{\boldsymbol{x}}^{\mathrm{syn}} \exp \left( -\sum_{k=1}^K \lambda_k q_{k,\boldsymbol{x}} \right).$$

Now we substitute $\mu^*$ into the Lagrangian function and get

$$L(\boldsymbol{p}^{\mathrm{post},*}, \boldsymbol{\lambda}^+, \boldsymbol{\lambda}^-, \mu^*) = -\log \sum_{\boldsymbol{x} \in \mathcal{X}} p_{\boldsymbol{x}}^{\mathrm{syn}} \exp \left( -\sum_{k=1}^K \lambda_k q_{k,\boldsymbol{x}} \right) - \sum_{k=1}^K a_k \lambda_k - \gamma \sum_{k=1}^K (\lambda_k^+ + \lambda_k^-).$$

Note that the $(\boldsymbol{\lambda}^+, \boldsymbol{\lambda}^-) \geq 0$ that maximizes $L(\boldsymbol{p}^{\text{post},*}, \boldsymbol{\lambda}^+, \boldsymbol{\lambda}^-, \mu^*)$ must satisfy $\lambda_k^+ \lambda_k^- = 0$ since, otherwise, we can replace them with

$$\left(\lambda_k^+ - \min\{\lambda_k^+, \lambda_k^-\}, \lambda_k^+ - \min\{\lambda_k^+, \lambda_k^-\}\right)$$

and the Lagrangian function increases. Hence, we have $\|\boldsymbol{\lambda}\|_1 = \sum_{k=1}^K \left(\lambda_k^{+,*} + \lambda_k^{-,*}\right)$. Now we simplify the Lagrangian function:

$$
\begin{aligned}
L(\boldsymbol{p}^{\text{post},*}, \boldsymbol{\lambda}^+, \boldsymbol{\lambda}^-, \mu^*) &= -\log \sum_{\boldsymbol{x} \in \mathcal{X}} p_{\boldsymbol{x}}^{\text{syn}} \exp\left(-\sum_{k=1}^K \lambda_k q_{k,\boldsymbol{x}}\right) - \sum_{k=1}^K a_k \lambda_k - \gamma\|\boldsymbol{\lambda}\|_1 \\
&= -\log\left(\mathbb{E}\left[\exp\left(-\sum_{k=1}^K \lambda_k q_k(\mathbf{X})\right)\right]\right) - \sum_{k=1}^K a_k \lambda_k - \gamma\|\boldsymbol{\lambda}\|_1 \\
&= -\log\left(\mathbb{E}\left[\exp\left(\sum_{k=1}^K -\lambda_k \left(q_k(\mathbf{X}) - a_k\right)\right)\right]\right) - \gamma\|\boldsymbol{\lambda}\|_1.
\end{aligned}
$$

Finally, we rewrite the optimal primal solution as

$$
\begin{aligned}
P^{\text{post},*}(\boldsymbol{x}) &\propto P^{\text{syn}}(\boldsymbol{x}) \exp\left(-\sum_{k=1}^K \lambda_k^* q_k(\boldsymbol{x})\right) \\
&\propto P^{\text{syn}}(\boldsymbol{x}) \exp\left(-\sum_{k=1}^K \lambda_k^* (q_k(\boldsymbol{x}) - a_k)\right).
\end{aligned}
$$

$\square$

## B  More Details on Algorithm Implementation

**Denoising output from Laplace mechanism.** For a given matrix $\boldsymbol{Q} \in \mathbb{R}^{K \times S}$ and a vector $\boldsymbol{a} \in \mathbb{R}^K$, we can denoise the outputs from Laplace mechanism by solving a linear program (LP):

$$
\begin{aligned}
\boldsymbol{p}^* &= \underset{\boldsymbol{p} \in \Delta_S^+}{\arg\min} \|\boldsymbol{Q}\boldsymbol{p} - \boldsymbol{a}\|_1 \\
&= \underset{\boldsymbol{p} \in \mathbb{R}^S}{\arg\min} \mathbf{1}^T \boldsymbol{t} \\
&\quad\quad \boldsymbol{Q}\boldsymbol{p} - \boldsymbol{a} \leq \boldsymbol{t} \\
&\quad\quad \boldsymbol{Q}\boldsymbol{p} - \boldsymbol{a} \geq -\boldsymbol{t} \\
&\quad\quad \mathbf{1}^T \boldsymbol{p} = 1 \\
&\quad\quad \boldsymbol{p} > 0.
\end{aligned}
$$

**Denoising output from Gaussian mechanism.** For a given matrix $\boldsymbol{Q} \in \mathbb{R}^{K \times S}$ and a vector $\boldsymbol{a} \in \mathbb{R}^K$, we can solve the following quadratic program to denoise the outputs from Gaussian mechanism:

$$
\boldsymbol{p}^* = \underset{\boldsymbol{p} \in \Delta_S^+}{\arg\min} \frac{1}{2}\|\boldsymbol{Q}\boldsymbol{p} - \boldsymbol{a}\|_2^2.
$$

**Implementing Algorithm 1 in log-domain.** Recall that we introduced a stochastic first-order method for computing the dual variables in Algorithm 1. To avoid underflow and overflow problems, we use log-domain computations to implement this algorithm. The details are included in Algorithm 2.

## C  Additional Experimental Results

In Table 3, we reproduce our experiment in Table 1 with a higher privacy budget. Our observation is consistent with prior experimental results: our post-processing technique consistently enhances the utility of synthetic data on selected metrics across all benchmark datasets and all privacy mechanisms.

**Algorithm 2** Dual variables computation in log scale.

**Input:** $\mathcal{D} = \{\bar{q}_i\}_{i=1}^n$ where $\bar{q}_i = (q_1(\boldsymbol{x}_i) - a_1, \cdots, q_K(\boldsymbol{x}_i) - a_K)$; maximal number of iterations $T$; mini-batch index $\mathcal{B}_t \subseteq [n]$; step size $\alpha_t$

**for** $t = 1, \cdots, T$ **do**

    Dual variables update (in log-scale):

$$ll_j^{(t+1)} = \exp\left( \text{prox}_{\alpha_t \gamma \|\cdot\|_1} \left( \log\left(ll_j^{(t)}\right) + \frac{\alpha_t}{\tau^{(t)}} \frac{1}{|\mathcal{B}_t|} \sum_{i \in \mathcal{B}_t} \prod_{k=1}^{K} ll_k^{(t)\left(-\bar{q}_{i,k}\right)} \bar{q}_{i,j} \right) \right) \quad \text{for } j \in [K]$$

$$\tau^{(t+1)} = \frac{1}{|\mathcal{B}_t|} \sum_{i \in \mathcal{B}_t} \prod_{k=1}^{K} ll_k^{(t+1)\left(-\bar{q}_{i,k}\right)}$$

**end for**

**Output:** $\boldsymbol{\lambda}^{(T+1)} = \log(\boldsymbol{ll}^{(T+1)})$

**Function:** $\text{prox}_{\alpha\|\cdot\|_1}(x) = \text{sign}(x) \max\{|x| - \alpha, 0\}$

| Dataset | DP Mechanism | Utility Improv. | F1 score | | JS distance (marginal) | | Inverse KL (marginal) | |
|---|---|---|---|---|---|---|---|---|
| | | | w/o post-proc. | w/ post-proc. | w/o post-proc. | w/ post-proc. | w/o post-proc. | w/ post-proc. |
| Adult | AIM | **0.29** $\pm0.04$ | 0.61 | 0.61 $\pm0.0$ | 0.01 | 0 $\pm0.0$ | 0.99 | 1 $\pm0.0$ |
| | MST | **0.1** $\pm0.09$ | 0.55 | 0.55 $\pm0.0$ | 0.01 | 0 $\pm0.0$ | 1 | 1 $\pm0.0$ |
| | DPCTGAN | **0.52** $\pm0.02$ | 0.56 | 0.58 $\pm0.03$ | 0.07 | 0.02 $\pm0.0$ | 0.81 | 0.87 $\pm0.0$ |
| | PATECTGAN | **0.7** $\pm0.01$ | 0.16 | 0.49 $\pm0.01$ | 0.02 | 0.01 $\pm0.0$ | 0.88 | 0.93 $\pm0.0$ |
| Mushroom | AIM | **0.28** $\pm0.0$ | 0.93 | 0.95 $\pm0.0$ | 0.01 | 0 $\pm0.01$ | 0.96 | 1 $\pm0.0$ |
| | MST | **0.55** $\pm0.0$ | 0.83 | 0.84 $\pm0.0$ | 0.01 | 0 $\pm0.0$ | 1 | 1 $\pm0.0$ |
| | DPCTGAN | **0.74** $\pm0.04$ | 0.61 | 0.73 $\pm0.02$ | 0.07 | 0.02 $\pm0.0$ | 0.72 | 0.85 $\pm0.0$ |
| | PATECTGAN | **0.8** $\pm0.09$ | 0.29 | 0.78 $\pm0.08$ | 0.02 | 0.01 $\pm0.0$ | 0.94 | 0.99 $\pm0.0$ |
| Shopper | AIM | **0.17** $\pm0.01$ | 0.5 | 0.5 $\pm0.0$ | 0.01 | 0.01 $\pm0.0$ | 0.89 | 0.97 $\pm0.0$ |
| | MST | **0.47** $\pm0.01$ | 0.47 | 0.52 $\pm0.0$ | 0.01 | 0 $\pm0.0$ | 0.98 | 1 $\pm0.0$ |
| | DPCTGAN | **0.1** $\pm0.09$ | 0.19 | 0.25 $\pm0.01$ | 0.06 | 0.02 $\pm0.01$ | 0.8 | 0.89 $\pm0.0$ |
| | PATECTGAN | **0.23** $\pm0.06$ | 0.22 | 0.26 $\pm0.05$ | 0.02 | 0.01 $\pm0.0$ | 0.83 | 0.96 $\pm0.02$ |
| Bank | AIM | **0.21** $\pm0.05$ | 0.47 | 0.47 $\pm0.0$ | 0.02 | 0.02 $\pm0.0$ | 0.93 | 0.98 $\pm0.0$ |
| | MST | **0.21** $\pm0.02$ | 0.41 | 0.44 $\pm0.0$ | 0.01 | 0 $\pm0.0$ | 0.98 | 1 $\pm0.0$ |
| | DPCTGAN | **0.15** $\pm0.03$ | 0.23 | 0.27 $\pm0.02$ | 0.03 | 0.01 $\pm0.0$ | 0.77 | 0.92 $\pm0.0$ |
| | PATECTGAN | **0.23** $\pm0.03$ | 0.16 | 0.19 $\pm0.05$ | 0.02 | 0.01 $\pm0.0$ | 0.94 | 0.97 $\pm0.01$ |

Table 3: We compare synthetic data generated without and with our post-processing technique, all under the same privacy budget $\epsilon = 4$.

