# OpenReview forum: "Post-processing Private Synthetic Data for Improving Utility on Selected Measures"
_NeurIPS.cc/2023/Conference — NeurIPS 2023 poster_

### Official Review · Reviewer_rPbc · 2023-06-18

**Soundness:** 4 excellent
**Presentation:** 4 excellent
**Contribution:** 3 good
**Rating:** 7
**Confidence:** 4

**Summary:**

The paper considers how to obtain improved utility from synthetic data under differential privacy.  Prior work has tried to achieve utility by incorporating utility directly into the generation of the synthetic data e.g., by making use of workload data.  Here, the proposal is to perform post-hoc reweighting of the synthetic data, based on measuring a difference in values between the final data under a set of queries with known answers.  This measurement/reweighting step is also done under privacy.  The paper shows that the optimization can be done using standard solvers, and so can be performed effectively via linear or quadratic programming (for pure or approximate DP). A specific use case based on aligning the correlation matrix is used to demonstrate that the sensitivity calculations can be quite simple.  Numerical experiments demonstrate that the approach can be quite effective in terms of absolute error ratio, and F1 score.

**Strengths:**

Synthetic data is widely promoted as a practical way to gain from private data, without running into issues around exhaustion of privacy budget.  Ensuring that synthetic data is usable for the target queries is therefore an important challenge, particularly if the workload may not be known at data generation time.  This approach shows a promising way to overcome these issues, provided we still have access to the data to perform the private optimization step.

The approach is very clean, and shows good promise in terms of utility and performance.

**Weaknesses:**

Evaluation is on four benchmark data sets that may be considered "easy".  It would be interesting to test this out on additional data, such as that used in recent synthetic data challenges by NIST.

**Questions:**

Apart from correlation matrix, can you generate more examples of query families to apply that have bounded sensitivity?

**Limitations:**

No need to address societal impact.

Technical limitations are outlined in Section 5 in broad strokes, but it would be interesting to hear more specific suggestions about how this approach could be extended to handle more diverse data.

---

> ### Author Rebuttal · Authors · 2023-08-10
>
> We thank the reviewer for their careful read of our paper and constructive comments!
>
> ---------
> **Q1. Evaluation is on four benchmark data sets that may be considered "easy". It would be interesting to test this out on additional data, such as that used in recent synthetic data challenges by NIST.**
>
> A1. To clarify, we have also applied our algorithm to the home-credit dataset [MOK18], in addition to the four benchmark datasets. The home-credit dataset is a large-scale dataset, which has 307,511 rows and 104 features. The execution of our method, which includes computing the utility measures from real data, denoising the noisy answers, and computing optimal resampling weights, on this large-scale dataset took approximately 3 mins on a single NVIDIA GeForce RTX 3090 GPU and the results are shown in Table 2 and Figure 1 in the paper.
>
>
> ---------
> **Q2. Apart from correlation matrix, can you generate more examples of query families to apply that have bounded sensitivity?**
>
> A2. Indeed, our proposed method can be expanded for use in many other applications, not limited to correlation alignment. For instance, it can be applied to mitigate data biases while ensuring the DP guarantees of resampled synthetic data. During the generation of private synthetic data, outliers may end up receiving a higher level of noise. This could disproportionately affect minority groups. As a result, ML models, when trained on this DP-synthetic data by downstream users, may exhibit disparate impact when implemented on real data.
>
> Specifically, consider a situation where each data record, represented by $x$, includes a sensitive attribute, such as gender or ethnicity, indicated by $s \in \mathcal{S}$. This record also includes an outcome variable, denoted as $y \in \mathcal{Y}$. To adjust the probability distribution $P_{S,Y}$ of the synthetic data to match that of the real data, our post-processing technique can be applied by selecting the following utility measures: $\set{q_{i,j}(x) = \mathbb{I}[s = i, y = j]}_{i\in \mathcal{S}, j \in \mathcal{Y}}$ where $\mathbb{I}$ is the indicator function.
>
> This method provides an extension to the current data pre-processing techniques found in fair ML literature [e.g., Kamiran and Calders, 2012] by implementing DP guarantees within the pre-processing pipeline. Broadly, our technique can be applied to enhance any utility measures that can be represented as bounded queries.
>
>
> —-Kamiran, F. and Calders, T., 2012. Data preprocessing techniques for classification without discrimination.
>
>
> ---------
> **Q3. Technical limitations are outlined in Section 5 in broad strokes, but it would be interesting to hear more specific suggestions about how this approach could be extended to handle more diverse data.**
>
> A3. Thank you for raising this important point! There are two additional applications where our approach can be effectively applied. First, in cases where the data exhibit a temporal structure, our approach can be used for post-processing synthetic data. This allows for the alignment of the transition matrix between adjacent time-steps with that of real data. The second application involves aligning the distribution of synthetic data in its tail with real data, as DP mechanisms tend to add higher noise to these regions. This aspect of preserving the tail distribution is particularly crucial in financial data, such as in fraud detection.

---

> > ### Comment · Reviewer_rPbc · 2023-08-11
> >
> > Thank you for these thorough responses to the questions and comments in the review.   Applications and evaluation in the context of fairness seems like an interesting direction to pursue.

---

> > > ### Author Response · Authors · 2023-08-11
> > > **Thank you for your prompt response!**
> > >
> > > Thank you for your prompt response! We are pleased to hear that you find the new applications and evaluations we presented in our response to be of interest. We will make sure to incorporate them into the revised paper. Finally, we would like to express our gratitude once more for the insightful and constructive comments you provided.

---

### Official Review · Reviewer_5bmq · 2023-06-21

**Soundness:** 3 good
**Presentation:** 2 fair
**Contribution:** 2 fair
**Rating:** 6
**Confidence:** 3

**Summary:**

The paper proposes a DP post-processing method for synthetic data that weights the synthetic data to match user-selected utility measures on the real dataset. The utility measures on real data are measured with noise to make this post-processing DP. To find the synthetic data weights, the paper derives a closed-form expression, and develops an optimisation algorithm to find optimal dual variables required to evaluate the closed-form optimum. To evaluate the method, the paper conducts experiments on 5 real datasets, testing how much the method is able to improve the synthetic data produced by 5 existing methods of generating DP synthetic data.


**Strengths:**

The paper is written well, and the main points are easy to understand. The idea behind the proposed method is interesting, and it should, at least in principle, be applicable to any kind of synthetic data and utility measure.


**Weaknesses:**

Some important experimental details are missing. See my questions.

The Private PGB method of Neunhoeffer et al. (2021) is fairly similar to the proposed method, and their differences should be discussed. Currently the paper is not even cited.

Minor comments:
- Slater's condition should be introduced before the proof of Theorem 1.
- Seeing the raw utility and F1 scores from the experiment in Table 1 would be useful, as it would show whether the large improvement from post-processing on the GANs is just caused by the GANs generating poor synthetic data
- References [App17], [Dat23], [DG17], [MOK18], [Sma23] should have URLs


**Questions:**

Important experimental details:
- Is the plain synthetic data baseline $\epsilon$ or $(\epsilon + \epsilon')$-DP? It seems to be the former, which is not a fair comparison, as the generation + post-processing is $(\epsilon + \epsilon')$-DP.
- Are the categorical columns also converted to numerical features for AIM and MST? They should not be, as both methods handle categorical values natively.
- What is the workload given to AIM? Does it contain the marginal queries with the variables the utility measures are looking at?

Minor questions:
- How is the real data split into training and test data?
- Why were MST and AIM not run on the home-credit dataset? The original paper on Private-PGM (McKenna et al. 2019) that both methods are based on experiments on a dataset with a similar number of features.
- Is there a difference between $\lambda^*$ in (4) and $\lambda$ in (5)?

References:
- Neunhoeffer et al. "Private Post-GAN Boosting" ICLR 2021
- McKenna et al. "Graphical-model based estimation and inference for differential privacy" ICML 2019


**Limitations:**

The paper mentions that they assume the real data is normalised so each feature lies in $[0, 1]$. This is a fairly large limitation, as normalising data under DP while retaining the possibility of undoing the normalisation is not trivial, and should be discussed further..

---

> ### Author Rebuttal · Authors · 2023-08-10
>
> We thank the reviewer for the thoughtful comments and for appreciating the value of the work!
>
> ---
> **Q1. Some important experimental details are missing.**
>
> A1. We appreciate your constructive feedback regarding our experimental results! Please find the detailed responses to your questions below, where we hope to adequately address all your concerns.
>
> ---
> **Q2. Private PGB method of Neunhoeffer et al. (2021).**
>
> A2. We thank the reviewer for pointing out the missing reference. We will ensure it is included in the revised paper. The key difference between our work and [Neunhoeffer etal., 2021] is that their method is tailored to GAN-based privacy mechanisms while our approach is model-agnostic. In other words, it can be applied to improve the utility of synthetic data generated by *any* privacy mechanisms. This versatility is crucial, given that marginal-based and workload-based mechanisms often yield higher quality synthetic data, as evidenced by benchmark experiments in [Tao etal., 2021], the NIST competition rankings [McKenna etal., 2021], and Table 1, 2 in the PDF submitted in our global response. Our experiments indicate that our method consistently improves the utility of synthetic data produced by all kinds of privacy mechanisms, even when the initial synthetic data is of high quality.
>
> —Tao, Y., McKenna, R., Hay, M., Machanavajjhala, A. and Miklau, G., 2021. Benchmarking differentially private synthetic data generation algorithms.
>
> —McKenna, R., Miklau, G. and Sheldon, D., 2021. Winning the NIST Contest: A scalable and general approach to differentially private synthetic data.
>
> ---
> **Q3. Slater's condition should be introduced before the proof of Theorem 1.**
>
> A3. We thank the reviewer for their suggestion and will include Slater's condition before the proof of Theorem 1.
>
> ---
> **Q4. Seeing the raw utility and F1 scores from the experiment in Table 1 would be useful, as it would show whether the large improvement from post-processing on the GANs is just caused by the GANs generating poor synthetic data**
>
> A4. Absolutely, that's an excellent suggestion! Please refer to the PDF we included in our global response for the raw F1 scores of both the original synthetic data and the post-processed data.
>
> ---
> **Q5. Some references should have URLs**
>
> A5. We appreciate the reviewer's suggestion and will incorporate the URLs into the revised paper.
>
> ---
> **Q6. Is the plain synthetic data baseline $\epsilon$ or $\epsilon + \epsilon’$-DP? It seems to be the former, which is not a fair comparison.**
>
> A6. In response to your concerns, we have conducted *an additional experiment* under the setup you suggested. Please refer to the PDF we submitted in the global response for details. In short, our observations are in line with prior experiments, confirming that our algorithm consistently enhances the utility of synthetic data on selected measures.
>
> The reasoning behind our original setup stems from the observation that increasing the privacy budget in synthetic data generation mechanisms *does not* necessarily enhance the quality of the synthetic data, even when averaged over multiple trials. We conjecture this incongruity arises from the random noise injected into the data generation process to ensure DP guarantees. Therefore, even without employing our post-processing algorithm, synthetic data generated from a lower privacy budget may surpass data produced with a higher privacy budget in terms of their overall quality. Our initial experimental setup enabled us to eliminate this potential issue. As a result, we concluded that our post-processing algorithm can consistently enhance the utility of the synthetic data on selected measures, despite requiring a small privacy budget (see lines 45–46 and 53–55).
>
> ---
> **Q7. Are the categorical columns also converted to numerical features for AIM and MST?**
>
> A7. To clarify, we used ordinal encoding to pre-process the categorical columns, which were then fed into the AIM and MST mechanisms.
>
> ---
> **Q8. Workload given to AIM**
>
> A8. The workload given to AIM consists of all one-way and two-way marginals. The utility measures chosen for our study are first-order and second-order moments, which are functions of the workload.
>
> ---
> **Q9. How is the real data split into training and test data?**
>
> A9. We split the real data, using 80% for generating synthetic data and setting aside 20% to evaluate the performance of predictive models trained on the synthetic data.
>
> ---
> **Q10. Why were MST and AIM not run on the home-credit dataset?**
>
> A10.  The primary reason was the lack of a functional GPU-accelerated implementation for these algorithms at the time of our experimentation. Our attempts to generate synthetic copies of the home-credit dataset, which comprises 100+ columns, using CPU-based computations proved to be exceedingly time-consuming and resource-intensive. Despite our best efforts, these attempts repeatedly led to kernel crashes and hindered our ability to carry out comprehensive experiments.
>
> ---
> **Q11. Difference between $\lambda^{*}$ in (4) and $\lambda$ in (5)?**
>
> A11. $\lambda$ represents the variables involved in the optimization problem in Eq. (5), while $\lambda^*$ denotes the optimal solution. We will clarify this notation in the updated version of our paper.
>
> ---
> **Q12. Data normalization.**
>
> A12. To clarify, our proposed method can be expanded to a broader context where we do not necessarily assume that each feature is in $[0,1]$. Instead, we only require utility measures that are represented by bounded queries. In the case of aligning the correlation matrix as an application, the only requirement is for all features to have a bounded domain, ensuring that the first and second-order moment queries are also bounded. Users can either define the lower and upper bounds of each feature, or they can be estimated from real data using a DP mechanism (see e.g., an OpenDP API: snsynth.transform.minmax).

---

> > ### Comment · Reviewer_5bmq · 2023-08-11
> >
> > Thank you for the response. You addressed my biggest concerns with the experimental setup very well, so I'm moving to recommend acceptance.

---

> > > ### Author Response · Authors · 2023-08-11
> > > **Thank you for your prompt response!**
> > >
> > > Thank you for your prompt response! We are glad to hear that our response has addressed your biggest concerns. We will ensure that the responses provided above, along with the promised changes, are integrated into the revised paper. Finally, we would like to express our gratitude once more for the insightful and constructive comments you provided.

---

### Official Review · Reviewer_zgZz · 2023-07-02

**Soundness:** 4 excellent
**Presentation:** 3 good
**Contribution:** 3 good
**Rating:** 7
**Confidence:** 4

**Summary:**

The paper introduces a post-processing technique that enhances the utility of synthetic DP data with respect to selected measures.
The proposed technique involves resampling from the synthetic data to filter out samples that do not meet the selected utility measures, using a stochastic first-order algorithm to find optimal resampling weights.

**Strengths:**

Strengths:

- The post-processing technique discussed appears to be novel and significant.

- The method is model-agnostic. This makes the technique highly versatile.

- The authors provide a good set of numerical experiments to validate their approach. These results are strong and show improvements across multiple benchmark datasets and synthetic data generation algorithms.

- The paper is well-structured and the methodology is clearly explained.


**Weaknesses:**

Weaknesses:

- The paper could benefit from a more detailed discussion on the trade-offs involved in using the proposed post-processing technique. For instance, it would be helpful to understand the impact of the technique on the computational complexity of the data synthesis process.

- The authors could provide more examples or case studies to illustrate the practical applications and benefits of their proposed technique, beside the correlation example illustrated.


**Questions:**

- Could you elaborate on the computational complexity of the proposed post-processing technique? How does it compare to the complexity of existing synthetic data generation algorithms?

- How does the proposed technique handle high-dimensional data? Are there any limitations or challenges in this regard?

- Could you provide additional examples to illustrate the practical applications and benefits of the proposed technique?

**Limitations:**

See my questions above.

---

> ### Author Rebuttal · Authors · 2023-08-10
>
> We thank the reviewer for the kind comments and the encouragement!
>
> ---------
> **Q1. More detailed discussion on the trade-offs involved in using the proposed post-processing technique and the impact of the technique on the computational complexity of the data synthesis process.**
>
> A1. That’s a great suggestion! Indeed, scalability is a key feature of our proposed technique. To be precise, our Algorithm 1 requires a computational cost of $O(bKT)$ where $b$ denotes the mini-batch size, $K$ denotes the number of utility measures of interest, and $T$ denotes the number of iterations. Note that this computational cost is independent with the number of features or the number of synthetic data. In contrast, some existing approaches do not scale to high-dimensional data. For example, they need to solve an integer program multiple times [Vietri etal., 2020] or need to solve a large-scale optimization problem whose complexity depends on the number of synthetic data to generate  [Aydore etal., 2021]. Please also refer to our response to your Q4 for further details on how our algorithm scales on a large-scale real-world dataset.
>
> —Vietri, G., Tian, G., Bun, M., Steinke, T. and Wu, S., 2020. New oracle-efficient algorithms for private synthetic data release.
>
> —Aydore, S., Brown, W., Kearns, M., Kenthapadi, K., Melis, L., Roth, A. and Siva, A.A., 2021. Differentially private query release through adaptive projection.
>
>
> ---------
> **Q2. Practical applications and benefits of their proposed technique, beside the correlation example illustrated.**
>
> A2. Thank you for your valuable suggestion. Yes, our proposed method can be expanded for use in many other applications, not limited to correlation alignment. For instance, it can be applied to mitigate data biases while ensuring the DP guarantees of resampled synthetic data.
> During the generation of private synthetic data, outliers may end up receiving a higher level of noise. This could disproportionately affect minority groups. As a result, ML models, when trained on this DP-synthetic data by downstream users, may exhibit disparate impact when implemented on real data.
>
> Specifically, consider a situation where each data record, represented by $x$, includes a sensitive attribute, such as gender or ethnicity, indicated by $s \in \mathcal{S}$. This record also includes an outcome variable, denoted as $y \in \mathcal{Y}$. To adjust the probability distribution $P_{S,Y}$ of the synthetic data to match that of the real data, our post-processing technique can be applied by selecting the following utility measures: $\set{q_{i,j}(x) = \mathbb{I}[s = i, y = j]}_{i\in \mathcal{S}, j \in \mathcal{Y}}$ where $\mathbb{I}$ is the indicator function.
>
> This method provides an extension to the current data pre-processing techniques found in fair ML literature [e.g., Kamiran and Calders, 2012] by implementing DP guarantees within the pre-processing pipeline. Broadly, our technique can be applied to enhance any utility measures that can be represented as bounded queries.
>
> —-Kamiran, F. and Calders, T., 2012. Data preprocessing techniques for classification without discrimination.
>
>
> ---------
> **Q3. The computational complexity of the proposed post-processing technique.**
>
> A3. Please refer to our response to your Q1.
>
> ---------
> **Q4. How does the proposed technique handle high-dimensional data? Are there any limitations or challenges in this regard?**
>
> A4. Indeed, a key characteristic of our proposed technique is its scalability, particularly with high-dimensional data. This scalability has been achieved by transforming the original optimization problem (Eq. 3), where the number of variables exponentially increases with the number of features, into a dual problem. Moreover, we have introduced a stochastic compositional proximal gradient algorithm (Algorithm 1) for solving the dual problem, enhancing computational efficiency significantly through the use of mini-batches for parameter updates. In addition, we have provided a PyTorch implementation of our algorithm which is optimized for GPU-based computations (please refer to the submitted code).
>
> As a practical demonstration, we have applied our proposed technique to the home-credit dataset [MOK18]. This dataset has 307,511 rows and 104 features. The execution of our method, which includes computing the utility measures from real data, denoising the noisy answers, and computing optimal resampling weights, on this large-scale dataset took approximately 3 mins on a single NVIDIA GeForce RTX 3090 GPU.
>
> ---------
> **Q5. Additional examples to illustrate the practical applications and benefits of the proposed technique.**
>
> A5. Please refer to our response to your Q2.

---

> > ### Comment · Reviewer_zgZz · 2023-08-11
> > **Re: rebuttal**
> >
> > Thank you for your responses to my questions! Including a discussion of the computational complexity of the method would indeed be helpful.

---

> > > ### Author Response · Authors · 2023-08-11
> > > **Thank you for your prompt response!**
> > >
> > > Thank you for your prompt response! Yes, we will make sure to include a detailed discussion about the computational complexity of our algorithm in the revised paper, along with a comparison to existing algorithms. Finally, we would like to express our gratitude once more for the insightful and constructive comments you provided.

---

### Official Review · Reviewer_SorM · 2023-07-05

**Soundness:** 2 fair
**Presentation:** 3 good
**Contribution:** 1 poor
**Rating:** 6
**Confidence:** 5

**Summary:**

The paper under review introduces a technique aimed at enhancing the quality of differentially private synthetic data. Specifically, this approach is applicable when a private synthetic data set, generated by any available privacy-preserving mechanism, does not align with the original data set on certain key measures or queries. The proposed solution involves adjusting the weights of synthetic data samples to ensure their alignment with the original data based on the specified objective. The paper specifically applies this technique to post-process synthetic data such that the resultant synthetic data aligns with the true data's correlation matrix.

The proposed algorithm commences by estimating the empirical correlation matrix using the Gaussian mechanism with privacy parameters epsilon=1 and epsilon=3. This involves the addition of independent Gaussian noise to each entry of the correlation matrix. Subsequently, the algorithm resolves a convex optimization problem using a first-order method to determine the optimal sample weights that best conform to the noisy correlation matrix.
The research findings demonstrate that the post-processed synthetic data offers a more accurate approximation of the correlation matrix compared to the original synthetic data (prior to the reweighting operation), thereby underscoring the effectiveness of the proposed method.


**Strengths:**

The idea of post-processing synthetic data presented in this paper is a compelling approach that boasts potential applicability to a multitude of problems. While the primary objective here is the alignment of pair-wise correlations within the data, theoretically, the objective could be a more intricate query that existing synthetic data methods are not designed to tackle.


**Weaknesses:**

In terms of originality, the proposed solution bears similarity to the Private Entropy Projection (PEP) mechanism referenced in [LVW21]. The PEP mechanism functions over the entirety of the data domain, assumed to be sufficiently small for its computational limitations. Nevertheless, it could, in theory, operate over any support, including samples from synthetic datasets, as suggested in this paper. Thus, it would be advantageous to explain the distinctions between these methods. Further, if they indeed differ, the paper should clarify why the PEP method is ill-suited for addressing the problem at hand.

Regarding the experimental setup, it appears to have some potential flaws. The paper compares the utility of a synthetic dataset D_syn, presumably generated using an original privacy budget (termed epsilon_original), with the utility of a post-processed dataset D_post, created with an additional privacy budget of epsilon=1. Consequently, the post-processed dataset D_post possesses less privacy than the pre-processed D_syn. Hence, it remains ambiguous whether the superiority of D_post over D_syn can be attributed to the proposed resampling method or merely to the fact that a larger privacy budget was utilized to generate D_post. A more transparent comparison would involve assessing the utility of D_post against a synthetic dataset generated using a budget of (epsilon_original + 1).

Lastly, while the post-processing operation enhances the synthetic data in relation to the correlation matrix, it remains uncertain whether this operation compromises other essential data properties. For instance, if the original synthetic data was trained on 3-way marginals, the post-processing step could cause a deviation from these queries, thus leading to a poorer approximation. This possibility could account for the machine learning results in Figure 1, where the post-processed dataset exhibits a lower F1 score under certain conditions.


**Questions:**

Why was the algorithm GEM only used for the home-credit dataset ?
What is the effect of the post-processes dataset on the measures that the original data was optimized for? That is, does the utility degrade?
How does the quality of the original synthetic data affect the absolute performance of the post-process synthetic data ?

**Limitations:**

Yes

---

> ### Author Rebuttal · Authors · 2023-08-10
>
> We thank the reviewer for the thoughtful review and for appreciating the merits of the work!
>
> ---------
> **Q1. Comparison with PEP in [LVW21].**
>
> A1. Thank you for highlighting this comparison. In contrast to PEP in [LVW21], our proposed solution offers several enhancements.
>
> First, it is important to note that their Algorithm 4, an implementation of PEP outlined in the appendix, only applies when $\gamma=0$. The extension of this algorithm to accommodate a general non-negative violation tolerance, i.e., $\gamma \geq 0$, is not straightforward. In contrast, we introduced a stochastic compositional proximal gradient algorithm (Algorithm 1 in our paper) that is applicable to any general non-negative violation tolerance. From our numerical experiments, we have observed that setting gamma to be a small positive number (for example, $\gamma = 1e-5$ in all our setups) consistently outperforms when $\gamma = 0$.
>
> Second, note that PEP is designed to find an optimal solution to a regularized constraint optimization problem (Eq. 8 in their appendix). However, due to the Laplace/Gaussian noise introduced to the measurements, this problem may prove to be infeasible. Despite the authors' claim in Appendix C.2 that their algorithm can still be executed even if Equation 8 is infeasible, solving the dual optimization under this circumstance may not yield a satisfactory primal solution. In contrast, our approach is to first denoise the noisy measurements prior to running our algorithm. By doing this, we can consistently guarantee the feasibility of the constraint optimization problem (Eq. 3 in our paper). Furthermore, this denoising step could augment the accuracy of our proposed solution since it can be perceived as identifying the maximum likelihood estimation of the utility measures of interest.
>
> Finally, we wish to highlight the originality of both the challenges we tackled in this study—incorporating end-user requirements into the data generation pipeline—and our proposed solution—a post-processing pipeline to enhance the utility of synthetic data. These subjects have not been explored in any previous research.
>
>
>
> ---------
> **Q2. Experimental setup appears to have some potential flaws.**
>
> A2. In response to your comments, we have conducted an additional experiment under the setup you suggested: comparing the utility of synthetic data produced by privacy mechanisms with a privacy budget of $\epsilon_{original} + 1$, against the post-processed synthetic data that was generated by applying our post-processing technique (with a privacy budget of $1$) to synthetic data generated from privacy mechanisms with a privacy budget of $\epsilon_{original}$.  Please refer to the PDF we submitted in the global response for more details.
>
>
> In short, our observations are in line with prior experiments, confirming that our algorithm consistently enhances the utility of synthetic data on selected measures.
>
> The reasoning behind our initial experimental setup stems from the observation that increasing the privacy budget in synthetic data generation mechanisms *does not* necessarily enhance the quality of the synthetic data, even when averaged over multiple trials. The metrics we considered in this quality assessment include utility measures of interest, downstream performance, and statistical measures. We conjecture this incongruity arises from the random noise injected into the data generation process to ensure DP guarantees. Therefore, even without employing our post-processing algorithm, synthetic data generated from a lower privacy budget may surpass data produced with a higher privacy budget in terms of their overall quality. Our initial experimental setup enabled us to eliminate this potential issue. As a result, we concluded that our post-processing algorithm can consistently enhance the utility of the synthetic data on selected measures, despite requiring a small privacy budget (see lines 45–46 and 53–55).
>
>
>
>
> ---------
> **Q3. Whether post-processing compromises other essential data properties.**
>
> A3. Indeed, this is a great point. In response to your concerns, we have incorporated two statistical metrics (average inverse of KL-divergence and Jensen-Shannon distance) along with the F-1 score to evaluate the quality of synthetic data. These metrics are implemented by using synthcity, which is a python package for generating and evaluating synthetic tabular data. As shown, our post-processing algorithm consistently enhances the selected utility measures without compromising either statistical parameters or downstream performance measures.
>
> ---------
> **Q4. GEM model and other experimental results**
>
> A4. Due to GEM not being included in the opendp package, we omitted it from Table 1 out of concern that comparing GEM with other mechanisms, processed under different data pre-processing pipelines, may lead to misleading conclusions.
>
> In response to your other comments, note that the workloads of AIM include all one-way and two-way marginals. The utility measures presented in Table 1 are based on first-order and second-order moments, which are functions of the workloads in AIM, and our post-processing algorithm continues to yield utility improvements across all datasets.
>
> We observed that the poorer the quality of the synthetic data, the greater the enhancement of quality through our post-processing algorithms. More specifically, we've seen that marginal-based mechanisms typically outperform DP-SGD-based mechanisms (please refer to the raw F1 scores reported in Table 1 and 2 in our global response). As shown in these two tables, our post-processing algorithms tend to yield more substantial utility enhancements for DP-SGD-based mechanisms.

---

> > ### Comment · Reviewer_SorM · 2023-08-15
> >
> > Thank you for your response. You have addressed my major concerns and I will move change my score to accept.

---

> > > ### Author Response · Authors · 2023-08-15
> > >
> > > Thank you so much for your response. We are glad to know that we have addressed your major concerns. We will make sure to include the promised changes in the revision (both in the main text and appendix).

---

### Author Rebuttal · Authors · 2023-08-10

We would like to thank all the reviewers for taking the time and effort to review our paper! We are delighted to learn that our paper was positively received, and the reviewers found that the idea of post-processing synthetic data presented in this paper is a compelling approach that boasts potential applicability to a multitude of problems (Reviewer SorM); the method is model-agnostic, which makes the technique highly versatile (Reviewer zgZz); the idea behind the proposed method is interesting, and it should, at least in principle, be applicable to any kind of synthetic data and utility measure(Reviewer 5bmq); and the approach is very clean, and shows good promise in terms of utility and performance (Reviewer rPbc). We also recognize that the reviewers are busy handling multiple papers, so their thoughtful feedback is even more appreciated.

---
We have included a PDF in this global response that contains additional experiments as suggested by the reviewers. The main differences compared with Table 1 in our submission are:

[New setup]. We compare the utility of synthetic data produced by privacy mechanisms with a privacy budget of $\epsilon + 1$, against the post-processed synthetic data that are generated by applying our post-processing technique (with a privacy budget of $1$) to synthetic data generated from privacy mechanisms with a privacy budget of $\epsilon$.

[Report raw F1 scores]. We report the F1 scores, both with and without our post-processing, instead of just the F1 score improvement. For your reference, the F1 scores of training on 80% real data and testing on 20% real data are: 0.61 on adult; 0.95 on mushroom; 0.54 on shopping; and 0.47 on bank.

[Include two statistical measures]. We include two statistical measures (Jensen-Shannon distance and average inverse of the KL-divergence) for evaluating statistical properties of synthetic data.

In short, our observations are in line with prior experiments, confirming that our algorithm consistently improves the utility of the synthetic data across all datasets and all DP mechanisms without degrading the performance of downstream models or statistical metrics.

---

Below, we address the concerns and questions raised by each reviewer and detail our plans for updating the paper. We will add the changes in the final version (both in the main text and appendix).

We welcome any additional feedback or suggestions that can further strengthen our paper and would be glad to hear from the reviewers.

Thanks!

---

### Decision · Program_Chairs · 2023-09-21

**Decision:**

Accept (poster)

**Comment:**

# Meta Review

The reviewers agree that the post-processing technique proposed in this paper is widely applicable and compelling, and most agree that the paper is well written. On the other hand, there are three main recurring weaknesses in the reviews:

First, reviewers SorM and 5bmq pointed out that the total privacy budget of the post-processed data uses a larger privacy budget, since the post-processing incurs additional privacy loss. During the discussion period the authors conducted additional experiments showing that this is not the source of improved performance, and the reviewers were convinced by these experiments. I think the new experiments resolve this concern and should be included in the paper.

Second, reviewers SorM and zgZz have concerns about trade-offs / compromises when using the proposed approach. For example, it is possible that post-processing the data to improve the correlation matrix may decrease the quality of the data in other ways. To some extent it seems like such trade-offs must be present, but I don't think that undermines the results much. The authors also conducted additional experiments showing that some specific statistical measures did not degrade after applying the post-processing technique.

Finally, during the reviewer discussion, reviewer SorM pointed out some additional baselines that could be good to compare against. I have included their comments below so that the authors may take them into account.

Overall, I think the reviewers are in agreement that the strengths outweigh the weaknesses and that the paper is above the bar for NeurIPS.

# Related work comments from reviewer SorM:

More importantly, It has become more apparent to me that there exists prior work capable of addressing the problem outlined in the paper. However, these approaches were not taken into consideration. Specifically, the previous work from [1] proposes a mechanism, referred to as PMW^{Pub}, that can tackle a problem very similar to the one in this paper. The only difference lies in the fact that the technique described in [1] conducts a post-processing procedure on publicly available data (as opposed to rows generated by an alternative synthetic data mechanism). Technically, PMW^{Pub} creates a support using a set of public records and subsequently employs the MWEM algorithm to reweigh the importance of each row in the support. In contrast, the support on this paper comes from rows of synthetic data. Regrettably, I initially overlooked this paper in my initial review. It does seem, though, that PMW^{Pub} could serve as a direct point of comparison within this paper.

Finally, I realized that the set of statistics (or workload) they optimize in this paper is very straightforward (The workload Q is defined under line 225). This means that recent generative mechanism such as GEM ([2]) or RAP ([3]) are equipped to handle these types of workloads. Notably, [2] shows how you can initialized GEM with public data. Therefore, a sensible baseline for the setting of the paper would be GEM initialized with synthetic data from other methods.

[1] Liu, Terrance, et al. "Leveraging public data for practical private query release." International Conference on Machine Learning. PMLR, 2021.

[2] Liu, Terrance, Giuseppe Vietri, and Steven Z. Wu. "Iterative methods for private synthetic data: Unifying framework and new methods." Advances in Neural Information Processing Systems 34 (2021): 690-702.

[3] Aydore, Sergul, et al. "Differentially private query release through adaptive projection." International Conference on Machine Learning. PMLR, 2021.